# Hybrid RSS/AOA Localization using Approximated Weighted Least Square in Wireless Sensor Networks

**DOI:** 10.3390/s20041159

**Published:** 2020-02-20

**Authors:** SeYoung Kang, TaeHyun Kim, WonZoo Chung

**Affiliations:** 1Division of Computer and Communications Engineering, Korea University, Seoul 02841, Korea; sykang0229@korea.ac.kr; 2Agency for Defense Development, Daejeon 34186, Korea

**Keywords:** wireless sensor network (WSN), target localization, weighted least square (WLS), received signal strength (RSS), angle of arrival (AOA)

## Abstract

We present a target localization method using an approximated error covariance matrix based weighted least squares (WLS) solution, which integrates received signal strength (RSS) and angle of arrival (AOA) data for wireless sensor networks. We approximated linear WLS errors via second-order Taylor approximation, and further approximated the error covariance matrix using a least-squares solution and the variance in measurement noise over the sensor nodes. The algorithm does not require any prior knowledge of the true target position or noise variance. Simulations validated the superior performance of our new method.

## 1. Introduction

Recently, localization using wireless sensor networks (WSNs) has gained considerable attention given the emerging development of services based on location awareness [1,2,3,4,5,6,7,8,9,10,11,12] A WSN is a group of spatially dispersed sensors (anchors) that monitor and record the physical condition of the environment. WSN localization usually involves range or angle measurement [2,3,4,5]. However, WSNs were not originally designed for target localization, and errors in the available data (received signal strength (RSS) or angle of arrival (AOA)) are a major concern; the perfect position information of anchor nodes may not be feasible. Some target localization techniques, such as the so-called range-free approaches like in [13,14,15,16], do not rely on RSS/AOA measurements, and roughly estimate the target location by exploiting the finite ranges of the sensors in the grid of abundant nodes. On the other hand, in the so-called range-based approach, RSS/AOA measurements are used to estimate the target location directly. The range-based localization assumes that each anchor node is able to have the positions of the node and RSS as well as the AOA of the received signals and, hence, theoretically, the exact location of the target is attainable. However, the errors in RSS/AOA measurements are severe, and sophisticated integration of the RSS and AOA measurements has been investigated [17,18,19,20,21,22,23]. The challenge of the range-based localization algorithms is to overcome the limitation of the noisy RSS/AOA measurements with a practical computational complexity.

Target position estimation using hybrid RSS/AOA data is an optimization problem of a non-convex system [9]. To actually solve the problem, a maximum likelihood (ML) estimator has been employed [17]. As the ML approach imposes an unacceptable computational burden, a simplified linearization of the non-convex problem has been considered and a straightforward least squares (LS) method for target localization has been proposed [17]. In an effort to develop a target estimation algorithm that is almost as effective as ML but has a low computational complexity (similar to that of the LS method), several non-convex optimization techniques, such as semidefinite programming (SDP) relaxation [20] and second-order cone programming (SOCP) relaxation [21], have been proposed. However, the computational complexity of these algorithms greatly exceeds that of the LS method. Several attempts have been made to applying weights to the simplified linear equations. In [21], a squared-range weighted least squares (SR-WLS) algorithm was proposed, in which the selected weighting was inversely proportional to the distance between an anchor and the target. In [22], using a spherical representation of the target, a closed-form LS equation was derived with selected weights based on the distance from the target (the WLS algorithm). In [23], a generalized least squares (GLS) approach using an error covariance matrix (the WLLS algorithm) was used, but it requires precise noise variance data that may not be available in practice. The performance comparison study in [24] reported that the WLS algorithm exhibits state-of-art performance, with a complexity comparable to that of the LS method but without the requirement for noise variance data. However, the distance-related weights may not be optimal, and the performance could be improved if an error covariance matrix was used. Unfortunately, the error covariance of the linear WLS equation is extremely difficult to compute and no closed-form expression is available.

Here, we present an elaborate weighting scheme based on the approximated error covariance matrix for the approximated linear equation of the WLS algorithm as a replacement for distance-based weights. The errors in the WLS linear equation are not independent Gaussian, and it is difficult to analyze the probability density function (PDF). Using a second-order Taylor approximation, we derived an approximate error covariance matrix assuming perfect knowledge of the true target position and the variance in Gaussian measurement noise. To compute the approximated error covariance matrix without any knowledge of the target position or noise, we replaced the true target position with an LS solution and the variance in measurement noise over the sensor nodes. The resulting algorithm, which we term the “error covariance WLS” (ECWLS) algorithm, outperforms the WLS with only a marginal increase in computational complexity, as confirmed by simulations. In summary, the main contribution of this paper is the presentation of a range-based hybrid RSS/AOA target localization algorithm that outperforms existing range-based state-of-the-art WLS without prior knowledge of the noise level.

The rest of the paper is organized as follows. In Section 2, the target localization problem when RSS and AOA measurements are used is introduced with related works, along with the existing solutions. Section 3 presents the new algorithm. We derived an approximated error covariance matrix using a second-order Taylor approximation and developed a practical matrix computation. In Section 4, we provide the computational complexity of all algorithms. In Section 5, we evaluate the performance of our algorithm via simulations and compare it to existing methods. Section 6 concludes the paper.

## 2. System Model and Related Works

### 2.1. System Model

Consider a wireless sensor network (WSN) with *N* sensor (anchor) nodes located at $a_{i}\;=\;{\left[a_{ix},a_{iy},\;a_{iz}\right]}^{T}\in R^{3}$ for i=1,⋯,N. Here, $x_{o}={\left[x_{ox},x_{oy},x_{oz}\right]}^{T}$ is the unknown target location and $d_{i}$, $\phi_{i}$, and $\alpha_{i}$ are the distance from the target ($d_{i}=\left||x_{o}-a_{i}|\right|$) and the associated azimuth and elevation angles, respectively (Figure 1). Each anchor obtains an RSS from the target, denoted as $P_{i}$ [25], and measures the AOA of the signal, specifically the azimuth angle ${\hat{\phi }}_{i}$ and the elevation angle ${\hat{\alpha }}_{i}$. Measurement error may occur in relation to the target location, $x_{o}$, as follows: (1)$$\begin{array}{c} \begin{array}{cc} {\hat{P}}_{i} & =P_{0}-10\gamma \log_{10}\frac{\left||x_{o}-a_{i}|\right|}{d_{0}}+n_{i}\\ {\hat{\phi }}_{i} & =\tan^{-1}\left(\frac{x_{oy}-a_{iy}}{x_{ox}-a_{ix}}\right)+m_{i}\\ {\hat{\alpha }}_{i} & =\cos^{-1}\left(\frac{x_{oz}-a_{iz}}{\left||x_{o}-a_{i}|\right|}\right)+v_{i} \end{array},\;\;\mathrm{for}\;i=1,\cdots ,N, \end{array}$$
where $P_{0}\left(dBm\right)$ is the calibrated (reference) received power at a reference distance $d_{0}\;\left(d_{i}\geq d_{0}\right)$ and γ is the path loss exponent (PLE). The measurement noises, $n_{i}∼N\left(0,\sigma_{n_{i}}^{2}\right)$, $m_{i}∼N\left(0,\sigma_{m_{i}}^{2}\right)$, and $v_{i}∼N\left(0,\sigma_{v_{i}}^{2}\right)$, are assumed to be independent white zero mean Gaussian noise.

### 2.2. Related Works

Using the Gaussian noise measurement model, the ML estimator for the target location, ${\hat{x}}_{ML}$, is derived as follows [17]:(2)$$\begin{array}{c} {\hat{x}}_{ML}=\arg\max_{x}\sum_{i}\ln P\left({\hat{P}}_{i},{\hat{\phi }}_{i},{\hat{\alpha }}_{i}|x_{o}\right)=\arg\min_{x}\sum_{i}f_{i}\left({\hat{P}}_{i},{\hat{\phi }}_{i},{\hat{\alpha }}_{i},x_{o}\right) \end{array}$$
where
(3)$$\begin{array}{c} \begin{array}{cc} f_{i}\left({\hat{P}}_{i},{\hat{\phi }}_{i},{\hat{\alpha }}_{i},x_{o}\right)= & \frac{1}{\sigma_{n_{i}}^{2}}{\left({\hat{P}}_{i}-P_{0}+10\gamma \log_{10}\frac{\left||x_{o}-a_{i}|\right|}{d_{0}}\right)}^{2}\\ \; & +\frac{1}{\sigma_{m_{i}}^{2}}{\left({\hat{\phi }}_{i}-\tan^{-1}\left(\frac{x_{oy}-a_{iy}}{x_{ox}-a_{ix}}\right)\right)}^{2}\\ \; & +\frac{1}{\sigma_{v_{i}}^{2}}{\left({\hat{\alpha }}_{i}-\cos^{-1}\left(\frac{x_{oz}-a_{iz}}{\left||x_{o}-a_{i}|\right|}\right)\right)}^{2} \end{array}\;\;\mathrm{for}\;i=1,\cdots ,N. \end{array}$$

In the absence of noise, the observation-target Equation (Equation 1) can be transformed into
(4)$$\begin{array}{c} \begin{array}{cc} \lambda_{i}u_{i}^{T}\left(x_{o}-a_{i}\right)-\eta d_{0} & =0\\ c_{i}^{T}\left(x_{o}-a_{i}\right) & =0\\ {\left(u_{i}\cos\left(\alpha_{i}\right)-1\right)}^{T}\left(x_{o}-a_{i}\right) & =0, \end{array} \end{array}$$
where $\lambda_{i}=10^{\frac{P_{i}}{10\gamma }},\;\eta =10^{\frac{P_{0}}{10\gamma }},\;c_{i}={\left[-\sin\left(\phi_{i}\right),\cos\left(\phi_{i}\right),\;0\right]}^{T}$, $1={\left[0,0,1\right]}^{T}$, and $u_{i}$ is the unit direction, $u_{i}={\left[\cos\left(\phi_{i}\right)\sin\left(\alpha_{i}\right),\sin\left(\phi_{i}\right)\sin\left(\alpha_{i}\right),\cos\left(\alpha_{i}\right)\right]}^{T}$. In the presence of noise, by approximating the true angles in $c_{i}$ and $u_{i}$ according to the observation angle as $c_{i}={\left[-\sin\left({\hat{\phi }}_{i}\right),\cos\left({\hat{\phi }}_{i}\right),\;0\right]}^{T}$ and $u_{i}={\left[\cos\left({\hat{\phi }}_{i}\right)\sin\left({\hat{\alpha }}_{i}\right),\sin\left({\hat{\phi }}_{i}\right)\sin\left({\hat{\alpha }}_{i}\right),\cos\left({\hat{\alpha }}_{i}\right)\right]}^{T}$, we obtain the following system of linear equations of the target location based on the observations proposed in [21].

(5)$$\begin{array}{c} \begin{array}{cc} {\hat{\lambda }}_{i}u_{i}^{T}\left(x_{o}-a_{i}\right)-\eta d_{0} & =\varepsilon_{i1}\\ c_{i}^{T}\left(x_{o}-a_{i}\right) & =\varepsilon_{i2}\\ {\left(\cos\left({\hat{\alpha }}_{i}\right)u_{i}-1^{T}\right)}^{T}\left(x_{o}-a_{i}\right) & =\varepsilon_{i3} \end{array},\;\mathrm{for}\;i=1\cdots N, \end{array}$$
where $\varepsilon_{ij}$ is the error term attributable to the Gaussian noise. In a matrix form
(6)$$\begin{array}{c} Ax_{o}-b=\varepsilon , \end{array}$$
where
(7)$$\begin{array}{c} A=\left[\begin{array}{c} {\hat{\lambda }}_{1}u_{1}^{T}\\ \vdots \\ {\hat{\lambda }}_{N}u_{N}^{T}\\ c_{1}^{T}\\ \vdots \\ c_{N}^{T}\\ {\left(\cos\left({\hat{\alpha }}_{1}\right)u_{1}-1\right)}^{T}\\ \vdots \\ {\left(\cos\left({\hat{\alpha }}_{N}\right)u_{N}-1\right)}^{T} \end{array}\right],\;b=\left[\begin{array}{c} {\hat{\lambda }}_{1}u_{1}^{T}a_{1}+\eta d_{0}\\ \vdots \\ {\hat{\lambda }}_{N}u_{N}^{T}a_{N}+\eta d_{0}\\ c_{1}^{T}a_{1}\\ \vdots \\ c_{N}^{T}a_{N}\\ {\left(\cos\left({\hat{\alpha }}_{1}\right)u_{1}-1\right)}^{T}a_{1}\\ \vdots \\ {\left(\cos\left({\hat{\alpha }}_{N}\right)u_{N}-1\right)}^{T}a_{N} \end{array}\right], \end{array}$$
and $\varepsilon ={\left[\begin{array}{ccccccccc} \varepsilon_{11} & \varepsilon_{21} & \cdots \varepsilon_{N1} & \varepsilon_{12} & \cdots & \varepsilon_{N2} & \varepsilon_{13} & \cdots & \varepsilon_{N3} \end{array}\right]}^{T}$. When ε is a zero-mean Gaussian random vector, the generalized LS solution [26], i.e.,
(8)$$\begin{array}{c} {\hat{x}}_{GLS}={\left(A^{T}W^{-1}A\right)}^{-1}A^{T}W^{-1}b, \end{array}$$
becomes an ML solution for (Equation 4), where W=Cov(ε|A). However, the PDF of ε is not Gaussian and is difficult to compute given the nonlinear transform from (Equation 1) to (Equation 4). In [22], an alternative weighting approach was proposed; *W* was replaced with the weights inversely proportional to the distance from the target: $W=I_{3}⊗\mathrm{diag}\left(w\right)$, $w=[w_{i}]$, where
(9)$$\begin{array}{c} w_{i}=1-\frac{d_{i}}{\sum_{j}d_{j}}. \end{array}$$

This WLS algorithm exhibits the best performance among the various localization algorithms based on hybrid RSS/AOA measurements [24].

On the other hand, the spherical coordinate system yields a direct estimation of the target position from the RSS and AOAs:(10)$$\begin{array}{c} \begin{array}{cc} {\hat{x}}_{x} & =a_{ix}+{\hat{d}}_{i}\cos\left({\hat{\phi }}_{i}\right)\sin\left({\hat{\alpha }}_{i}\right),\\ {\hat{x}}_{y} & =a_{iy}+{\hat{d}}_{i}\sin\left({\hat{\phi }}_{i}\right)\sin\left({\hat{\alpha }}_{i}\right),\\ {\hat{x}}_{z} & =a_{iz}+{\hat{d}}_{i}\cos\left({\hat{\alpha }}_{i}\right), \end{array}\;\mathrm{for}\;i=1,\cdots ,N. \end{array}$$

The LS solution described above (Equation 10) is termed the LS algorithm [17]. The error covariance matrix for LS is simple to compute (see Appendix A, which corrects the missing terms in the correlation matrix presented in [24]). The weighted LS algorithm using the covariance matrix is termed the WLLS algorithm [23]. Although the LS method is intuitive and simple, the performance comparisons in [24] and in Section 6 in this paper showed that the WLS method outperforms the LS method, while the WLS method is, in turn, outperformed by the WLLS algorithm due to the introduction of error covariance weights.

## 3. The Proposed Method

In this section, we develop an approximated weighted least squares approach for WLS by (i) computing an approximated error covariance matrix using second-order Taylor approximation, (ii) approximating the true target information in the covariance matrix to that of the LS solution, and (iii) approximating the noise variance to the sample noise variance over anchors. Figure 2 shows the overall flowchart of the proposed method.

### 3.1. Approximated Covariance Matrix

The errors in (Equation 5) contain nonlinear transforms of observations that are the sums of the true parameters and noise, as the straightforward evaluation of (Equation 5) shows: (11)$$\begin{array}{c} \begin{array}{cc} \varepsilon_{i1}= & {\hat{\lambda }}_{i}\cos\left({\hat{\varphi }}_{i}\right)\sin\left({\hat{\alpha }}_{i}\right)\left({\hat{x}}_{x}-a_{ix}\right)+{\hat{\lambda }}_{i}\sin\left({\hat{\varphi }}_{i}\right)\sin\left({\hat{\alpha }}_{i}\right)\left({\hat{x}}_{y}-a_{iy}\right)\\ \; & +{\hat{\lambda }}_{i}\cos^{2}\left({\hat{\alpha }}_{i}\right)\left({\hat{x}}_{z}-a_{iz}\right)-\hat{\eta }d_{0},\\ \varepsilon_{i2}= & -\sin\left({\hat{\varphi }}_{i}\right)\left({\hat{x}}_{x}-a_{ix}\right)+\cos\left({\hat{\varphi }}_{i}\right)\left({\hat{x}}_{y}-a_{iy}\right),\\ \varepsilon_{i3}= & \frac{1}{2}\sin\left(2{\hat{\alpha }}_{i}\right)\cos\left({\hat{\varphi }}_{i}\right)\left({\hat{x}}_{x}-a_{ix}\right)+\frac{1}{2}\sin\left(2{\hat{\alpha }}_{i}\right)\sin\left({\hat{\varphi }}_{i}\right)\left({\hat{x}}_{y}-a_{iy}\right)\\ \; & +\sin^{2}\left({\hat{\alpha }}_{i}\right)\left({\hat{x}}_{z}-a_{iz}\right). \end{array} \end{array}$$

Assuming that the noise is small enough, we consider a second-order Taylor approximation of the nonlinear terms of observations as follows: (12)$$\begin{array}{cc} {\hat{\lambda }}_{i} & =10^{\frac{P_{i}+n_{i}}{10\gamma }}=\lambda_{i}10^{\frac{n_{i}}{10\gamma }}\approx \lambda_{i}\left(1+\bar{\gamma }n_{i}+\frac{1}{2}{\bar{\gamma }}^{2}n_{i}^{2}\right), \end{array}$$(13)$$\begin{array}{cc} \sin(\hat{\phi }) & =\sin\left(\phi_{i}+m_{i}\right)\approx \sin\left(\phi_{i}\right)+\cos\left(\phi_{i}\right)m_{i}-\frac{1}{2}\sin\left(\phi \right)m_{i}^{2}, \end{array}$$(14)$$\begin{array}{cc} \cos(\hat{\alpha }) & =\cos\left(\alpha_{i}+v_{i}\right)\approx \cos\left(\alpha_{i}\right)-\sin\left(\alpha_{i}\right)v_{i}-\frac{1}{2}\cos\left(\alpha_{i}\right)v_{i}^{2}, \end{array}$$
where $\lambda_{i}:=10^{\frac{P_{i}}{10\gamma }}$ is the true power exponent and $\bar{\gamma }:=\frac{\ln\left(10\right)}{10\gamma }$. The approximated errors, denoted by ${\hat{\varepsilon }}_{ij}$, are derived directly from (Equation 11) by using (Equation 12), (Equation 13), and (Equation 14) as follows:(15)$$\begin{array}{c} \begin{array}{cc} {\hat{\varepsilon }}_{i1}= & \eta d_{0}\bar{\gamma }n_{i}+\lambda_{i}\left(r_{i}\cos\left(\alpha_{i}\right)-\sin\left(\alpha_{i}\right)d_{iz}\right)v_{i}+\frac{1}{2}\eta d_{0}{\bar{\gamma }}^{2}n_{i}^{2}-\frac{1}{2}\lambda_{i}r_{i}\sin\left(\alpha_{i}\right)m_{i}^{2}\\ \; & -\frac{1}{2}\lambda_{i}\left(r_{i}\sin\left(\alpha_{i}\right)+\cos\left(\alpha_{i}\right)d_{iz}\right)v_{i}^{2}\\ {\hat{\varepsilon }}_{i2}= & -r_{i}m_{i}\\ {\hat{\varepsilon }}_{i3}= & \left(\cos\left(2\alpha_{i}\right)r_{i}-\sin\left(2\alpha_{i}\right)d_{iz}\right)v_{i}-\frac{1}{4}\sin\left(2\alpha_{i}\right)r_{i}m_{i}^{2}-\left(\sin\left(2\alpha_{i}\right)r_{i}-\cos\left(2\alpha_{i}\right)d_{iz}\right)v_{i}^{2}\\ \; & +\frac{1}{2}\sin\left(2\alpha_{i}\right)r_{i}-\sin^{2}\left(2\alpha_{i}\right)d_{iz}, \end{array} \end{array}$$
where $d_{iz}:=x_{oz}-a_{iz},$$r_{i}:=\sqrt{{\left(x_{ox}-a_{ix}\right)}^{2}+{\left(x_{oy}-a_{iy}\right)}^{2}}$.

The covariance of $\left\{\hat{\varepsilon_{ij}}\right\}$ has the following property:(16)$$\begin{array}{c} \mathrm{Cov}\left({\hat{\varepsilon }}_{ik},{\hat{\varepsilon }}_{jk^{\prime }}\right)=E\left(\left({\hat{\varepsilon }}_{ik}-E\left[{\hat{\varepsilon }}_{ik}\right]\right)\left({\hat{\varepsilon }}_{jk^{\prime }}-E\left[{\hat{\varepsilon }}_{jk^{\prime }}\right]\right)\right)=0, \end{array}$$
for i≠j or k=2, given the independence of $n_{i}$, $m_{i}$, and $v_{i}$, where E(·) denotes the expectation operator. A straightforward computation yields non-zero covariances:(17)$$\begin{array}{c} \begin{array}{cc} \mathrm{Var}({\hat{\varepsilon }}_{i1}) & =E({\left({\hat{\varepsilon }}_{i1}-E\left[{\hat{\varepsilon }}_{i1}\right]\right)}^{2})\\ \; & ={\left(\eta d_{0}\bar{\gamma }\right)}^{2}\sigma_{n_{i}}^{2}+{\left(\lambda_{i}\left(r_{i}\cos\left(\alpha_{i}\right)-\sin\left(\alpha_{i}\right)d_{iz}\right)\right)}^{2}\sigma_{v_{i}}^{2}+\frac{1}{2}{\left(\eta d_{0}{\bar{\gamma }}^{2}\right)}^{2}\sigma_{n_{i}}^{4}\\ \; & +\frac{1}{2}{\left(\lambda_{i}r_{i}\sin\left(\alpha_{i}\right)\right)}^{2}\sigma_{m_{i}}^{4}+\frac{1}{2}{\left(\lambda_{i}\left(r_{i}\sin\left(\alpha_{i}\right)+\cos\left(\alpha_{i}\right)d_{iz}\right)\right)}^{2}\sigma_{v_{i}}^{4} \end{array} \end{array}$$(18)$$\begin{array}{cc} \mathrm{Var}({\hat{\varepsilon }}_{i2}) & =E\left({\left({\hat{\varepsilon }}_{i2}-E\left[{\hat{\varepsilon }}_{i2}\right]\right)}^{2}\right)=r_{i}^{2}\sigma_{m_{i}}^{2} \end{array}$$(19)$$\begin{array}{c} \begin{array}{cc} \mathrm{Var}({\hat{\varepsilon }}_{i3}) & =E({\left({\hat{\varepsilon }}_{i2}-E\left[{\hat{\varepsilon }}_{i2}\right]\right)}^{2})\\ \; & ={\left(\cos\left(2\alpha_{i}\right)r_{i}-\sin\left(2\alpha_{i}\right)d_{iz}\right)}^{2}\sigma_{v_{i}}^{2}+\frac{1}{8}{\left(\sin\left(2\alpha_{i}\right)r_{i}\right)}^{2}\sigma_{m_{i}}^{4}\\ \; & +2{\left(\sin\left(2\alpha_{i}\right)r_{i}-\cos\left(2\alpha_{i}\right)d_{iz}\right)}^{2}\sigma_{v_{i}}^{4} \end{array} \end{array}$$(20)$$\begin{array}{c} \begin{array}{cc} \mathrm{Cov}({\hat{\varepsilon }}_{i1},{\hat{\varepsilon }}_{i3}) & =E\left(\left({\hat{\varepsilon }}_{i1}-E\left[{\hat{\varepsilon }}_{i1}\right]\right)\left({\hat{\varepsilon }}_{i3}-E\left[{\hat{\varepsilon }}_{i3}\right]\right)\right)\\ \; & =\left(\lambda_{i}\left(r_{i}\cos\left(\alpha_{i}\right)-\sin\left(\alpha_{i}\right)d_{iz}\right)\right)\left(\cos\left(2\alpha_{i}\right)r_{i}-\sin\left(2\alpha_{i}\right)d_{iz}\right)\sigma_{v_{i}}^{2}\\ \; & +\frac{1}{4}\lambda_{i}r_{i}^{2}\sin\left(\alpha_{i}\right)\sin\left(2\alpha_{i}\right)\sigma_{m_{i}}^{4}\\ \; & +\lambda_{i}\left(r_{i}\sin\left(\alpha_{i}\right)+\cos\left(\alpha_{i}\right)d_{iz}\right)\left(\sin\left(2\alpha_{i}\right)r_{i}-\cos\left(2\alpha_{i}\right)d_{iz}\right)\sigma_{v_{i}}^{4}. \end{array} \end{array}$$

Let $C(x_{o},\sigma_{m_{i}},\sigma_{v_{i}},\sigma_{n_{i}})$ denote the covariance matrix of $\hat{\varepsilon }={\left[\begin{array}{ccccccccc} {\hat{\varepsilon }}_{11} & {\hat{\varepsilon }}_{21} & \cdots {\hat{\varepsilon }}_{N1} & {\hat{\varepsilon }}_{12} & \cdots & {\hat{\varepsilon }}_{N2} & {\hat{\varepsilon }}_{13} & \cdots & {\hat{\varepsilon }}_{N3} \end{array}\right]}^{T}$ given the true target location $x_{o}$ and noise level $\left\{\sigma_{m_{i}}\},\{\sigma_{v_{i}}\},\{\sigma_{n_{i}}\right\}$. Each entry of *C* is given above, i.e., $C_{ij}=\mathrm{Cov}\left({\hat{\varepsilon }}_{i}{\hat{\varepsilon }}_{j}\right)$. Note that, when computing *C*, the exact target position $x_{o}$ and noise variances are required. Below, we discuss approximation of *C* in the absence of prior knowledge of the true target position and noise variances.

### 3.2. Computation of Approximated Covariance Matrix

To compute *C*, we replace the true target position with the LS solution of (Equation 5),
(21)$$\begin{array}{c} x_{LS\_WLS}={\left(A^{T}A\right)}^{-1}A^{T}b, \end{array}$$
which is termed the LS_WLS solution, and is equivalent to the LS solution in (Equation 4) and does not require any prior knowledge of the true target position or noise level. The simulations in Section 6 show that the LS_WLS method already outperforms the LS algorithm. Using this approximation, we obtain: (22)$$\begin{array}{c} \begin{array}{cc} P_{i} & =P_{0}-10\gamma \log_{10}\frac{\left||x_{LS\_WLS}-a_{i}|\right|}{d_{0}},\\ \phi_{i} & =\tan^{-1}\left(\frac{x_{LS\_WLS_{y}}-a_{iy}}{x_{LS\_WLS_{x}}-a_{ix}}\right),\\ \alpha_{i} & =\cos^{-1}\left(\frac{x_{LS\_WLS_{z}}-a_{iz}}{\left||x_{LS\_WLS}-a_{i}|\right|}\right), \end{array}\;\mathrm{for}\;i=1,\cdots ,N. \end{array}$$

All secondary constants are approximated as follows: (23)$$\begin{array}{c} \begin{array}{cc} \lambda_{i} & =10^{\frac{P_{i}}{10\gamma }}=10^{\frac{P_{0}-10\gamma \log_{10}\left(||x_{LS\_WLS}-a_{i}||/d_{0}\right)}{10\gamma }},\\ \eta d_{0} & =\lambda_{i}||x_{LS\_WLS}-a_{i}||,\\ d_{iz} & =x_{LS\_WLS_{z}}-a_{iz},\\ r_{i} & =\sqrt{{\left(x_{LS\_WLS_{x}}-a_{ix}\right)}^{2}+{\left(x_{LS\_WLS_{y}}-a_{iy}\right)}^{2}}. \end{array} \end{array}$$

Precise knowledge of the noise variances $\sigma_{n_{i}}^{2}$, $\sigma_{m_{i}}^{2}$, and $\sigma_{v_{i}}^{2}$ is essential when computing *C*. Although sample variance over time should be used to estimate noise variance, we use sample variance over anchors to avoid undesirable delays and the complexity associated with sample accumulation; we assume that the anchors (i.e., sensors) in the network are homogeneous: (24)$$\begin{array}{c} \begin{array}{cc} {\hat{\sigma }}_{n_{i}}^{2} & =\frac{1}{N}\sum_{i=1}^{N}{\left({\hat{P}}_{i}-P_{0}+10\gamma \log_{10}\frac{\left||x_{LS\_WLS}-a_{i}|\right|}{d_{0}}\right)}^{2}\\ {\hat{\sigma }}_{m_{i}}^{2} & =\frac{1}{N}\sum_{i=1}^{N}{\left({\hat{\phi }}_{i}-\tan^{-1}\left(\frac{x_{LS\_WLS_{y}}-a_{iy}}{x_{LS\_WLS_{x}}-a_{ix}}\right)\right)}^{2}\\ {\hat{\sigma }}_{v_{i}}^{2} & =\frac{1}{N}\sum_{i=1}^{N}{\left({\hat{\alpha }}_{i}-\cos^{-1}\left(\frac{x_{LS\_WLS_{z}}-a_{iz}}{\left||x_{LS\_WLS}-a_{i}|\right|}\right)\right)}^{2}. \end{array}, \end{array}$$

Finally, let $\bar{C}=C\left(\bar{x},\{{\hat{\sigma }}_{m_{i}}\},\{{\hat{\sigma }}_{v_{i}}\},\{{\hat{\sigma }}_{n_{i}}\}\right)$ denote the approximated covariance matrix computed using (Equation 22)–(Equation 24). The estimated target position becomes:(25)$$\begin{array}{c} x_{ECWLS}={\left(A^{T}{\bar{C}}^{-1}A\right)}^{-1}A^{T}{\bar{C}}^{-1}b. \end{array}$$

Our algorithm does not require any prior knowledge of the true target position or noise level, but approximates the generalized LS outcome with a complexity of O(N). Simulations show that our ECWLS method outperforms the WLS method and shows robust performance, even when each anchor node shows different noise variances and the number of such nodes is small.

We improve the performance of the WLS method by using approximate error covariance and estimating the variance of noise. The existing WLLS method assumed that the variance of noise is known. However, we can localize the target without noise levels.

## 4. Complexity Analysis

The desired complexity for a localization algorithm in a WSN is O(N), where *N* is the number of anchors. The proposed algorithm requires computation of a least square solution ${\hat{x}}_{LS\_WLS}$ in (Equation 21), approximated errors, an approximated error covariance matrix $\bar{C}$, estimations of noise variance, and a computation of a weighted least square $x_{ECWLS}$ in (Equation 25). All of the listed processes basically have O(N) complexity, except the computations involving $\bar{C}$, which may require $O(N^{2})$ complexity. In particular, computation of $x_{ECWLS}$ requires inversion of a 3N×3N matrix, while computation of $x_{LS\_WLS_{x}}$ does inversion of a 3×3 matrix. However, since $\bar{C}$ has the following sparse structure,
(26)$$\begin{array}{c} \bar{C}=\left[\begin{array}{ccc} D_{11} & 0 & D_{13}\\ 0 & D_{22} & 0\\ D_{31} & 0 & D_{33} \end{array}\right] \end{array}$$
where $D_{11},D_{22},D_{33},D_{31},D_{13}$ are diagonal matrices,
(27)$$\begin{array}{c} D_{11}=diag\left(\mathrm{Var}\left({\hat{\varepsilon }}_{i1}\right)\right),D_{22}=diag\left(\mathrm{Var}\left({\hat{\varepsilon }}_{i2}\right)\right),D_{33}=diag\left(\mathrm{Var}\left({\hat{\varepsilon }}_{i3}\right)\right),D_{13}=D_{31}=diag\left(\mathrm{Cov}\left({\hat{\varepsilon }}_{i1},{\hat{\varepsilon }}_{i3}\right)\right), \end{array}$$
building up $\bar{C}$ and computing its inversion takes only O(N);
(28)$$\begin{array}{c} {\bar{C}}^{-1}=\left[\begin{array}{ccc} Q^{-1} & 0 & -Q^{-1}D_{13}D_{33}\\ 0 & D_{22}^{-1} & 0\\ -Q^{-1}D_{31}D_{33} & 0 & D_{33}^{-1}+D_{33}^{-1}D_{13}Q^{-1}D_{31}D_{33}^{-1} \end{array}\right], \end{array}$$
where
(29)$$\begin{array}{c} Q=D_{11}-D_{33}^{-1}D_{13}. \end{array}$$

Therefore, the total computational complexity of the proposed algorithm is O(N). Table 1 summarizes the complexities of the existing algorithms.

Table 1 shows the computational complexities of all algorithms. The computational complexities of all algorithms depend on the network size, *N*. This feature is common to methods operating in a centralized manner [5], since all information is transferred to a central node (processor). From Table 1, we can see that the computational complexity of the proposed method is the same as that of the LS method.

## 5. Performance Results

In this section, we verify the performance of our method in numerical simulations that use the experimental settings employed in most previous studies. The target and anchors are randomly deployed inside a box with an edge length *B* for each Monte Carlo run. The reference distance, $d_{0}$, is set to 1 m, and the reference power, $P_{0}$, is assumed to be −10 dBm. The PLE changes according to the environmental conditions and is considered to exhibit a uniform distribution within the interval [2.2,2.8] for each trial (i.e., $\gamma_{i}∼\mathrm{Unif}\left(2.2,2.8\right)$ for i=1,⋯,N, as in [22]). Performance is evaluated by calculating the root mean square error (RMSE), defined as $RMSE=\sqrt{\frac{1}{M_{c}}\sum_{i=1}^{M_{c}}{\left||x_{oi}-{\hat{x}}_{i}|\right|}^{2}}$, where $M_{c}$ is the run number and ${\hat{x}}_{i}$ is the estimate of the true target location, $x_{oi}$, during the $i^{th}$ run.

We compare the performance of our algorithm to that of the LS method in [17], WLLS algorithm in [23], and LS_WLS and WLS method in [22]. We also evaluate the performance of the real ECWLS algorithm, which replaces the estimated noise with the true noise variances in the ECWLS algorithm. Note that the real ECWLS should have better performance than the ECWLS.

Figure 3 shows the RMSE according to the number of anchor nodes, *N*, for the edge length B=15 m. The noise level is set to $\sigma_{n}=6$ (dB), $\sigma_{m_{i}}=\sigma_{v_{i}}=10$ (deg), and $\gamma_{i}∼\mathrm{Unif}\left[2.2,2.8\right]$, $P_{0}=-10$ (dBm), $d_{0}=1$ m, and $M_{c}$ = 50,000. The simulations show that the LS_WLS method outperforms the LS method. The WLLS method, which uses true noise variance, is markedly superior to the LS method, while the WLS method is only slightly better than LS_WLS method and the WLLS method outperforms the WLS method. Furthermore, the proposed ECWLS method that uses estimated noise variance is better than the WLLS method, which uses true noise variance. Finally, the real ECWLS, the ECWLS with true noise variance that is not available in practice, shows the best performance.

The RMSE of the proposed algorithm was investigated under conditions of inhomogeneous noise variance among the anchor nodes. The noise variances in each node were not constant, but were assumed to have a uniform distribution. Figure 4a shows a case with slight deviation in noise variance. The noise variances are set to $\sigma_{n_{i}}∼\mathrm{Unif}\left[3,9\right]$ (dB), $\sigma_{m_{i}}∼\mathrm{Unif}\left[6,12\right]$, and $\sigma_{v_{i}}∼\mathrm{Unif}\left[6,12\right]$ (deg). The other settings are the same as in the previous experiment, i.e., B=15 m, $\gamma_{i}∼\mathrm{Unif}\left[2.2,2.8\right]$, $P_{0}=-10$ (dBm), $d_{0}=1$ m, and $M_{c}=50000$. Figure 4b increases the differences in noise variance. The noise variances are set to $\sigma_{n_{i}}=\mathrm{Unif}\left[1,10\right]$ (dB), $\sigma_{m_{i}}∼\mathrm{Unif}\left[1,20\right]$, and $\sigma_{v_{i}}∼\mathrm{Unif}\left[1,20\right]$ (deg).

In both plots, the proposed ECWLS algorithm outperforms the WLLS algorithm (which requires the exact noise variance of all anchor nodes), even in the severe noise estimation error due to the deviation of noise variances among nodes. The proposed ECWLS algorithm is outperformed only by the real ECWLS, which requires the true noise variance data, as clearly illustrated by Figure 4b.

Figure 5 compares the RMSE of the proposed ECWLS algorithm with those of the LS and WLS algorithms at various noise levels for a fixed anchor number N=4. The settings in Figure 5a are $\sigma_{m_{i}}=10$ (deg), $\sigma_{v_{i}}=10$ (deg), $\gamma_{i}∼\mathrm{Unif}\left[2.2,2.8\right]$, $P_{0}=-10$ (dBm), $d_{0}=1$ m, and $M_{c}=$ 50,000. Figure 5a shows that the LS method is sensitive to RSS noise, but the WLS and ECWLS methods are not. The settings in Figure 5b are $\sigma_{n_{i}}=6$ (dB), $\sigma_{v_{i}}=10$ (deg), $\gamma_{i}∼\mathrm{Unif}\left[2.2,2.8\right]$, $P_{0}=-10$ (dBm), $d_{0}=1$ m, and $M_{c}$ = 50,000. The settings in Figure 5c are $\sigma_{n_{i}}=6$ (dB), $\sigma_{m_{i}}=10$ (deg), $\gamma_{i}∼\mathrm{Unif}\left[2.2,2.8\right]$, $P_{0}=-10$ (dBm), $d_{0}=1$ m, and $M_{c}=50000$. For various noise levels, the proposed method performed better than all of the other methods.

Finally, we evaluate the performance of the proposed algorithm on a large scale. Figure 6 shows the RMSE according to the number of anchor nodes, *N*, for the edge length B=150 m. The noise level is set to $\sigma_{n}=6$ (dB), $\sigma_{m_{i}}=\sigma_{v_{i}}=10$ (deg), and $\gamma_{i}∼\mathrm{Unif}\left[2.2,2.8\right]$, $P_{0}=-10$ (dBm), $d_{0}=1$ m, and $M_{c}=50000$. The proposed method performed better than all of the other methods on a large scale.

## 6. Conclusions

We present a novel target localization algorithm based on hybrid RSS/AOA measurements, termed the ECWLS algorithm. The distance-based weights of the WLS algorithm are replaced by an approximated error covariance matrix that can be computed easily without prior knowledge of the noise level. The superior performance of our ECWLS algorithm was confirmed by numerical simulations.

## Figures and Tables

**Figure 1 sensors-20-01159-f001:**
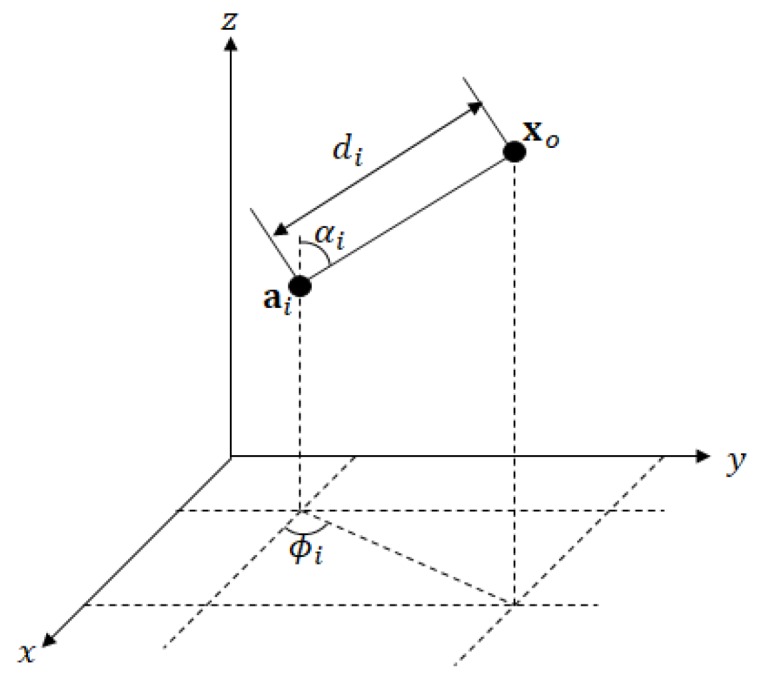
Anchor node and target location in a 3-D space.

**Figure 2 sensors-20-01159-f002:**
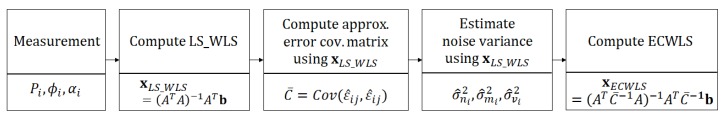
The flowchart of the error covariance weighted least squares (ECWLS) algorithm.

**Figure 3 sensors-20-01159-f003:**
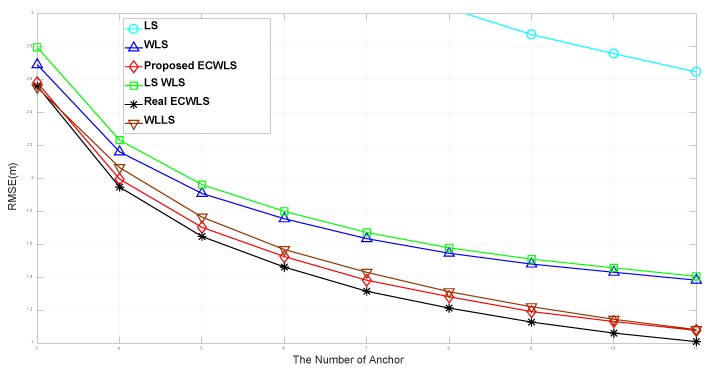
Root mean square error (RMSE) vs. the number of anchor nodes.

**Figure 4 sensors-20-01159-f004:**
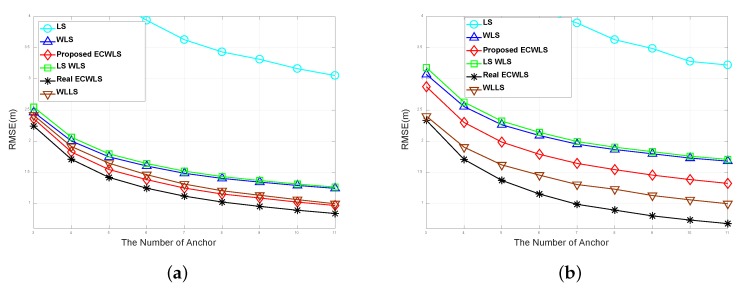
RMSE vs. the number of anchor nodes under noise dispersion with (**a**) slight differences in noise variance and (**b**) large differences in noise variance.

**Figure 5 sensors-20-01159-f005:**
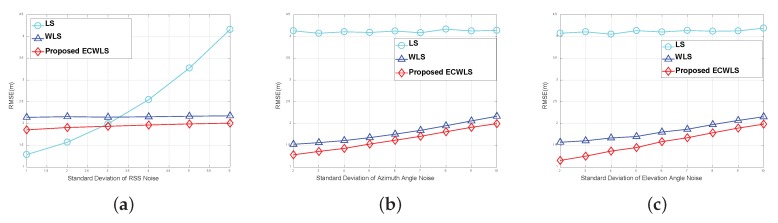
(**a**) RMSE vs. standard deviation of received signal strength (RSS) noise; (**b**) RMSE vs. standard deviation of azimuth angle noise; (**c**) RMSE vs. standard deviation of elevation angle noise.

**Figure 6 sensors-20-01159-f006:**
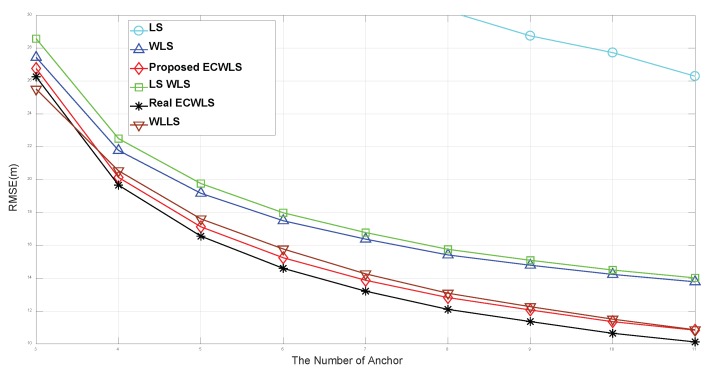
RMSE vs. the number of anchor nodes on a large scale.

**Table 1 sensors-20-01159-t001:** Complexities of all algorithms.

Algorithm	Description	Complexity
LS	The LS method in [17]	O(N)
LS_WLS	The LS method in [22]	O(N)
WLS	The WLS method in [22]	O(N)
WLLS	The WLLS method in [23]	O(N)
ECWLS	The proposed ECWLS method	O(N)

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
