# Peer review of "Hybrid RSS/AOA Localization using Approximated Weighted Least Square in Wireless Sensor Networks"

_sensors, 2020, doi:10.3390/s20041159_

Round 1
Reviewer 1 Report
Please describe better the difference between Proposed ECWLS and Real ECWLS. Add more comments to measured results.
Reviewer 2 Report
The manuscript presents a method for localization of targets which is based on weighted least squares solution of system of linear equations. As weight function is used approximated error covariance matrix. Numerical experiments validate proposed new method.
Additional comments:
Instead of "linear equation" write "system of linear equations" before (5). It is not clear how (14) are obtained. Write more details with references to used equations. Page 5, row 92: Reformulate "Note that ...". Now this sentence is not clear and E is not explained. Explain with more details and references to used equations how (16)-(18) are obtained. Equation (11): not clear, check it or explain more why you don't have derivatives of lambda in the second and third terms of Taylor's expression.Author Response
Please see the attachment.

Reviewer 4 Report
Authors present a target localization method using an approximated error covariance matrix based weighted least squares (WLS) solution, which integrates received signal strength (RSS) and angle of arrival (AoA) data for wireless sensor networks.
The article describes very clearly the motivation and fit of their method in the state of the art, the development of it and the evaluation of its performance through simulations.
In my opinion, the paper almost can be accepted in this form. Just some very minor comments:
- Remove the reference [16] in line 3 from the abstract.
- The references [2] and [8] are the same.
- On line 134, the words "our method" are repeated.
- The quality of the figures showing the results of the simulations can be improved: on the one hand, increase the size of the markers to make them easier to read when the paper is printed in black and white. On the other hand, include the figures in vector format, not bitmap.
Round 2
Reviewer 3 Report
The authors have addressed all my concerns.